# Complement C3 Regulates Inflammatory Response and Monocyte/Macrophage Phagocytosis of *Streptococcus agalactiae* in a Teleost Fish

**DOI:** 10.3390/ijms232415586

**Published:** 2022-12-09

**Authors:** Hao Bai, Liangliang Mu, Li Qiu, Nuo Chen, Jiadong Li, Qingliang Zeng, Xiaoxue Yin, Jianmin Ye

**Affiliations:** 1Guangdong Provincial Key Laboratory for Healthy and Safe Aquaculture, Guangzhou Key Laboratory of Subtropical Biodiversity and Biomonitoring, School of Life Sciences, South China Normal University, Guangzhou 510631, China; 2Guangdong Laboratory for Lingnan Modern Agriculture, Guangzhou 510642, China

**Keywords:** *Oreochromis niloticus*, complement 3 (C3), *Streptococcus agalactiae*, inflammatory response, phagocytosis

## Abstract

The complement system is composed of a complex protein network and is pivotal to innate immunity. Complement 3 (C3) is a critical protein in the complement cascade and participates in complement activation and immune defense. In this study, C3 from Nile tilapia (*Oreochromis niloticus*) was cloned and its function in resisting pathogen infection was characterized. The full length of OnC3 open reading frame is 4974 bp, encoding 1657 aa, and the predicted protein mass weight is 185.93 kDa. The OnC3 amino acid sequence contains macroglobulin domains. The expression pattern of OnC3 mRNA in the tissues of healthy fish was detected, with the highest in the liver and the lowest in the muscle. After challenged with *Streptococcus agalactiae* and *Aeromonas hydrophila*, the expression of OnC3 mRNA was significantly up-regulated in the liver, spleen, and head kidney. Further, the recombinant OnC3 protein alleviated the inflammatory response and pathological damage of tissues after infected with *S. agalactiae*. Moreover, the OnC3 promoted the phagocytosis of monocytes/macrophages to *S. agalactiae*. The data obtained in this study provide a theoretical reference for in-depth understanding of C3 in host defense against bacterial infection and the immunomodulatory roles in teleost fish.

## 1. Introduction

Complement, a vital constituent of the innate immune system, is a complex protein network composed of more than 50 kinds of proteins and protein sub-fragments [1,2,3,4]. The activation of complement is achieved through the classical pathway, lectin pathway, as well as alternative pathway [3]. The classical pathway is activated by the binding of complement C1q to antigen-complexed IgM (CH3 domain) or IgG (CH2 domain) [5]. The lectin pathway is launched by the binding of mannose-binding lectin (MBL), collectins, and ficolins to carbohydrates present on the surface of pathogens [6]. In addition, the alternative pathway is activated by viruses, bacteria, and fungi, and directly activates C3 by passing C1, C2, and C4 [7]. Although the initiator molecules and components involved in the three activation pathways are different, they all center on proteolytic activation of C3. Then, the downstream C5–C9 proteins are activated, leading to the complement cascade effects and formation of the membrane-attack complex (MAC) [1,8]. After being activated, the complement system induces chemotaxis and cell activation by producing anaphylatoxins and mediates inflammation by promoting phagocytosis, degranulation, and cell lysis. The crucial immunological functions of the complement system include host defense against pathogens, elimination of immune complexes, and promotion of adaptive immune responses [9,10]. All those pathways will converge in complement component 3 (C3), the core factor of the complement system, which plays a crucial role in innate and adaptive immune response [1,8,11].

C3 is a *β*2 glycoprotein composed of *α* and *β* peptide chains and mainly synthesized by the liver [12,13]. The concentration of C3 is stabilized at 1.0–1.5 g/L in healthy human serum [13,14]. When stimulated by irritants such as pathogenic bacteria, the content of C3 in serum is up-regulated and cleaved by C3 convertase to participate in complement activation. The C3 is cleaved into small C3a fragments and large C3b fragments [10]. Both C3a and C3b combine with the complement receptors on the surface of the cell membrane to perform immunomodulatory functions, including C3a receptors (C3aR) and C3 receptors 1–4 (CR1–CR4) [15,16,17]. C3a is an anaphylatoxin that modulates the production of inflammatory factors such as *IL-1* and stimulates respiratory burst such as macrophages and the chemotaxis of mast cells [18,19]. C3b is an opsonin that promotes phagocytosis of macrophages and immunoadhesion of erythrocytes [20,21]. Interestingly, C3b is then cleaved into fragments of different sizes, such as C3c, C3d, and C3g, which perform specific functions in the complement system or other immune mechanisms [10]. In addition, most mammalian models demonstrate that C3 plays a crucial role in the molecular pathogenesis of many different diseases, such as atypical hemolytic uremia and glomerular inflammation [22,23].

The sequence information and functions of C3 have been successfully characterized in several bony fishes including the grass carp (*Ctenopharyngodon idella*) [24,25], Japanese flounder (*Paralichthys olivaceus*) [26], large yellow croaker (*Larimichthys crocea*) [27], dojo loach (*Misgurnus anguillicaudatus*) [28], and rainbow trout (*Oncorhynchus mykiss*) [29]. In grass carp, complement C3 is involved in the process of resisting *A. hydrophila* and repairing tissue damage caused by bacterial infection [25]. In addition, C3a is involved in promoting the phagocytosis of phagocytic B cells through C3aR [24]. In Japanese flounder, PoC3 is found to have direct bactericidal effects, as well as chemotaxis and antibacterial infective effects [26]. These studies illustrate the functions of C3 in innate immunity of host defense. However, studies on C3 in phagocytosis and antibacterial immunity of bony fish are rarely reported, especially in Nile tilapia. Nile tilapia is the second most bred fish in the world and is important for the aquaculture industry of China [30]. The *S. agalactiae* and *A. hydrophila* infection causes serious damage to Nile tilapia tissues and affects the aquaculture industry negatively [31,32]. Therefore, further research on the mechanisms of Nile tilapia immune defense against pathogenic microorganisms is an urgent need for green aquaculture [8,30,31,32].

In the present study, the open reading frame (ORF) of the Nile tilapia complement C3 gene (*OnC3*) was successfully cloned, and the expression pattern after *S. agalactiae* and *A. hydrophila* stimulation was also confirmed. Furthermore, the recombinant OnC3 protein was expressed, and it was discovered that it could resist *S. agalactiae* infection by mediating inflammation and opsonizing the phagocytosis of monocytes/macrophages (MO/Mø). Moreover, (r)OnC3 was found to be able to reduce the tissue damage induced by *S. agalactiae* infection. Above all, this study proposes a new reference for understanding the functional properties of C3 in teleost host defense against pathogen infection.

## 2. Results

### 2.1. Sequence Analysis of OnC3

The ORF of *OnC3* is 4974 base pairs (bp) in length, encoding 1657 amino acids (aa). Different domain sequence information was marked with different colors according to the prediction of SMART (Appendix A). The deduced amino acid sequence of OnC3 shared 40.88%, 40.46%, 40.04%, 40.52%, 41.89%, and 95.11% identities with *Homo sapiens*, *Mus musculus*, *Bos taurus*, *Gallus*, *Xenopus laevis*, and *Maylandia zebra* (Figure 1A). The phylogenetic tree showed that *OnC3* is relatively conserved with C3 family members in bony fishes and is most homologous with that in *Maylandia zebra* (Figure 1B). In the tertiary image of *OnC3*, the predicted structure and relative position of different domains are visually presented, such as A2M_N, A2M_N_2, ANATO, A2M, A2M_comp, A2M_recep, and C345C (Figure 1C).

### 2.2. Tissue Distribution and Expression Pattern of OnC3

The expression of *OnC3* transcripts in healthy Nile tilapia illustrated that *OnC3* was extensively expressed in various tissues. The *OnC3* was expressed in the liver most abundantly and sequentially decreased in the brain, head kidney, thymus, spleen, skin, intestines, hind kidney, peripheral blood, and muscle. Significantly, the expression in the liver was about 62,000-fold of that in muscle (Figure 2A).

The expression patterns of *OnC3* in vivo and in vitro were significantly changed after *S. agalactiae* and *A. hydrophila* stimulation. In the liver, the *OnC3* transcripts raised massively and reached the maximum at 24 h after challenge. The expression of *OnC3* was 6-fold and 4-fold higher than the PBS group after challenged with *S. agalactiae* and *A. hydrophila*, respectively (Figure 2B). A similar tendency was observed in the spleen, as the expression of *OnC3* was 10-fold and 7-fold higher after stimulated by these two pathogenic bacteria (Figure 2C). However, the transcripts of *OnC3* in the head kidney fluctuated between 2-fold and 4-fold higher than the PBS group after stimulation (Figure 2D). In MO/Mø, the *OnC3* expression was 7-fold and 4-fold higher than that of the PBS group at 24 h after challenge with *S. agalactiae* and *A. hydrophila* (Figure 2E).

### 2.3. Effects of (r)OnC3 on Inflammatory Response In Vitro and In Vivo

The Ni-chelating affinity chromatography was applied, purifying the recombinant OnC3 protein. The SDS-PAGE was used to identify a band near 70 kDa, approaching the theoretical mass weight of (r)OnC3 (Appendix A).

To detect the effects of (r)OnC3 on inflammatory response, the expression of pro-inflammatory cytokines *IL-1β*, *IL-6*, *TNF-α* and anti-inflammatory cytokines *IL-10* and *TGF-β* transcripts in the liver and spleen were detected. The results illustrated that (r)OnC3 upregulated the transcripts of *IL-1β*, *IL-6*, *TNF-α,* and *TGF-β* in the liver and spleen at 12 h after stimulation (Figure 3A).

In MO/Mø, the expressions of all inflammatory cytokines were remarkably augmented after challenge with the (r)OnC3 protein. In the (r)OnC3 groups, the transcription of *IL-1β*, *IL-6*, *TNF-α,* and *IL-10* was about 5-fold to 7-fold higher than the Trx groups and PBS groups at 12 h, 24 h, and 48 h after stimulation (Figure 4A–D). The expression of *TGF-β* in the (r)OnC3 group also was 5-fold higher than that in other groups at 12 h after stimulation (Figure 4E).

### 2.4. (r)OnC3 Regulates ROS Levels in MO/Mø

Before the experiment, the cells were loaded with the DCFH-DA fluorescent probe, as the fluorescence intensity of DCF can represent the content of ROS. The ROS in MO/Mø was assayed by flow cytometric analysis. The results showed that the DCF intensity in the (r)OnC3 group was 60.6%, which was apparently stronger than the PBS group (50.9%) and Trx group (51.4%) (Figure 4F). The blank control group was 1.1%. Furthermore, the positive cell rate of the (r)OnC3 group was the highest (Figure 4G).

### 2.5. (r)OnC3 Regulates Inflammation after S. agalactiae Infection

According to the results of the in vivo experiment, the transcripts of *IL-1β*, *IL-6*, *TNF-α, IL-10,* and *TGF-β* in experimental groups were apparently higher than the PBS group after 12 h of stimulation. However, in the *S. agalactiae* + (r)OnC3 group, the transcripts of pro-inflammatory cytokines *IL-1β*, *IL-6*, and *TNF-α* were lower than the *S. agalactiae* group and *S. agalactiae* + Trx group, while the expression of *IL-10* and *TGF-β* was augmented compared to the *S. agalactiae* group and *S. agalactiae* + Trx group (Figure 5A,B).

Meanwhile, the expression of inflammation factors confirmed a similar changing tendency in MO/Mø (Figure 5C). In the experimental groups, all detected inflammatory factors were remarkably increased at 12 h after stimulation, and there were no apparent divergences between the *S. agalactiae* group and *S. agalactiae* + Trx group. In the *S. agalactiae* + (r)OnC3 stimulated group, the expression of *IL-1β*, *IL-6*, and *TNF-α* was lower than the *S. agalactiae* and *S. agalactiae* + Trx stimulated group, while the expression of *IL-10* and *TGF-β* was higher than the two groups.

### 2.6. (r)OnC3 Alleviates Inflammatory Response and Pathological Damage

The results of the histopathological analysis indicated that (r)OnC3 significantly induced the inflammatory infiltration of the liver and spleen (Figure 3B) and greatly increased the injury score in healthy fish (Figure 3C). When *S. agalactiae* were infected, the (r)OnC3 significantly alleviated the inflammatory response and injury score (Figure 5D,E). The histopathological analysis by H&E staining indicated that the inflammatory infiltration and hemosiderosis could be detected in liver and spleen of *S. agalactiae* and the *S. agalactiae* + Trx group. However, the pathological injury was greatly relieved in the *S. agalactiae* + (r)OnC3 group (Figure 5D). Additionally, the injury scores of the *S. agalactiae* + (r)OnC3 group were obviously lower in *S. agalactiae* and the *S. agalactiae* + Trx group (Figure 5E).

### 2.7. (r)OnC3 Regulates Monocyte/Macrophage Phagocytosis

To further examine the effect of (r)OnC3 in phagocytosis, the expressions of phagocytosis-related receptors (*CD11b*, *CD18*), enzymes (*RhoA*, *Rock*), as well as lysosome-related factors (*EEA*, *M6PR*, and *vATPase*) were detected by RT-qPCR. The results showed that in head kidney tissue, the transcripts of these factors in *S. agalactiae* + (r)OnC3 groups were significantly higher than in *S. agalactiae*, the *S. agalactiae* + Trx group, and the PBS group (Figure 6A). Analogously, results in MO/Mø showed that the expression of these factors in the *S. agalactiae* + (r)OnC3 group was higher than *S. agalactiae*, the *S. agalactiae* + Trx group, and the PBS group (Figure 6B). Furthermore, the assay of phagocytosis was developed with flow cytometric analysis (Figure 6C). The MO/Mø without phagocytosis showed that the phagocytic rate was only 1.6%. The phagocytic rate of MO/Mø to *S. agalactiae* in the (r)OnC3 group was 61.5%, much higher than the 48.0% and 51.0% in the PBS and Trx groups (Figure 6C,D).

## 3. Discussion

As a crucial constituent of innate immunity, the complement system has a long evolutionary history. Among them, C3, as an evolutionarily conserved core molecule, is ubiquitous in vertebrates and invertebrates [4,33]. C3 and its cleavage fragments bind to receptors on the cell surface, playing a variety of immunological roles, such as promoting phagocytosis and killing potential pathogens, or regulating the activation of the complement system for immune function [1,2,3,8]. Nonetheless, the specific immune mechanism of C3 is still unclear in early vertebrates. In this study, the molecular cloning and characterization of Nile tilapia C3 were performed. The role of OnC3 in defense against pathogenic bacteria from the perspective of phagocytosis and inflammatory responses was explored as well.

The results of this study revealed that OnC3 is composed of *α* and *β* chains, connected by disulfide bonds, and the putative cleavage site was located at 659–662 (RKKR). The OnC3 were predicted conserved with seven domains: A2M_N, A2M_N_2, ANATO, A2M, A2M_comp, A2M_recep, and C345C domains. Moreover, a highly conserved thioester region (GCGEQ) in *OnC3* was also predicted. The phylogenetic analysis indicated that *OnC3* clustered with C3 of other bony fish, and also has high similarity with mammals and invertebrates, which further reflects the high conservation of C3 in the evolutionary process [10,28,34].

In mammals, the liver is considered as the foremost organ of C3 synthesis [13]. Consistent with most reported teleosts, the expression of *OnC3* was the highest in the liver tissue of Nile tilapia [25,28]. The brain tissue was second only to the liver. Coincidentally, recent studies have also shown that C3 has a greater relationship with the homeostasis of the central nervous system [35]. Furthermore, the expression of *OnC3* mRNA in the head kidney, thymus, and spleen is also at a high level. After *S. agalactiae* and *A. hydrophila* infection, the expression levels of *OnC3* mRNA in the liver, spleen, and head kidney were significantly up-regulated. These organs with high expression of *OnC3* are important immune organs of Nile tilapia in antibacterial infection and are also suitable pathological models in vivo [2]. C3 is mainly synthesized in hepatocytes, and it is also synthesized in extrahepatic cells, including monocytes/macrophages, neutrophils, and fibroblasts [36]. Monocytes/macrophages are professional phagocytic cells and an important cellular model for studying phagocytosis and antibacterial immunity in vitro [37]. Similar to tissue challenge results in vivo, the transcription of *OnC3* mRNA in head kidney MO/Mø remarkably rose after infected by the two pathogens as well. The *S. agalactiae* and *A. hydrophila* are currently the main pathogens of Nile tilapia, causing huge economic losses to the aquaculture industry [31,32]. These results indicated that the expression of *OnC3* would increase after pathogen infection and that it is involved in the process of resisting pathogen infection, which further helps to explore the important mechanism of OnC3 in Nile tilapia resisting pathogen infection.

Complement components are present in an inactive state to prevent the negative effects of complement imbalances under normal circumstances. The imbalances of the complement activation are the important causes of tissue damage and end organ damage in autoimmune diseases, such as systemic lupus erythematosus and C3 glomerulopathy in humans [20,21,38]. To explore the effects of excess C3 on fish, healthy Nile tilapia were injected with the (r)OnC3 protein and the head kidney MO/Mø were stimulated with the (r)OnC3 protein. The expression of inflammatory factors (*IL-1β*, *IL-6*, *TNF-α*, *IL-10,* and *TGF-β*) were apparently ascended after injection of the (r)OnC3 protein in liver and spleen of healthy fish. The inflammatory factors and intracellular reactive oxygen species that increased significantly were also up-regulated in head kidney MO/Mø after (r)OnC3 challenge. Meanwhile, slight tissue damage was detected in liver and spleen by H&E staining. These results suggested the content and function of C3 in Nile tilapia is tightly regulated, preventing damage to the immune system due to the imbalance of complement [20].

The complement components will be activated and function in turn when stimulated [1,4]. After pathogenic bacteria invade the host, they can adapt to the host environment and proliferate rapidly and attack vital tissues such as spleen and liver through blood circulation [39]. The toxic substances released by these pathogenic bacteria can trigger an inflammatory response, resulting in different degrees of tissue damage [39,40]. The occurrence of inflammatory response is an obbligato process for the body to resist pathogenic infection. However, the damage caused by prolonged inflammatory response is also not negligible [41,42]. After infection with *S. agalactiae*, the liver and spleen of Nile tilapia were congested, with swelling and severe water accumulation [40,43,44,45]. A large number of vacuoles appeared in the liver tissue, and a large number of cell necrosis and inflammatory cells appeared in liver and spleen. The corresponding symptoms are significantly relieved by injecting an appropriate amount of the (r)OnC3 protein. Necrotic cells were reduced in the liver, the vacuolation phenomenon was alleviated, and there was no significant inflammatory infiltrate in the spleen. Furthermore, we found that (r)OnC3 could regulate the inflammatory response by inhibiting the expression of pro-inflammatory cytokines (*IL-6*, *IL-1β,* and *TNF-α*) and promoting the expression of anti-inflammatory cytokines (*IL-10* and *TGF-β*), thereby slowing down the tissue damage caused by *S. agalactiae*. These results suggested that C3 plays a vital role in host immune defense and has antibacterial effects against pathogen infection in bony fishes.

Phagocytic cells are chief components of innate immunity and act as a bridge between innate and adaptive immunity [37]. In mammals, activation of C3 is capable of affecting the phagocytosis of phagocytic cells such as macrophages or neutrophils [4]. Recent studies on pearl oysters pointed out that disrupting the *C3* gene could significantly reduce the phagocytosis of *Vibrio alginolyticus* by blood cells, which also showed the importance of C3 in invertebrate phagocytes [34]. Phagocytes are also the first line of defense in the innate immunity of bony fishes. Recent studies pointed out that C3a enhances the phagocytic activity of B cells through C3aR in grass carp [24,37]. The fragments of C3, such as C3b and iC3b, are extremely important in the complement system for opsonophagocytosis [15,16,46]. These fragments bind to complement receptors (CR) on the surface of macrophages, thereby affecting the phagocytic activity of macrophages, including CR1-CR4 [15]. CR3 (CD11b/CD18) is known as the principal *β*2 integrin, which plays a prominent role in recognizing bacteria. It can bind to inactivated C3b (iC3b) and activate phagocytosis [47,48]. Despite the signaling pathway(s) responsible for its cytokine response having not been scientifically elucidated, related studies also have reported that CR3-mediated phagocytosis requires the participation of RhoA and ROCK [49,50]. The Rho/ROCK signaling pathway acts on inducing cytoskeletal reorganization, cell migration, and stress fiber formation, thereby affecting tissue permeability and tissue contraction, and is also associated with phagosome formation [51,52,53]. In this study, (r)OnC3 significantly enhanced the phagocytosis of *S. agalactiae* by Nile tilapia head kidney MO/Mø. The expression of CR3-mediated phagocytosis-related genes was detected as well. Notably, *CR3* (*CD11b*/*CD18*), *RhoA,* and *ROCK* were all up-regulated. Significant up-regulation also occurred in association with phagolysosome formation such as *EEA*, *M6PR,* and *vATPase* [54]. These data suggested that OnC3 can effectively promote the phagocytosis of monocytes/macrophages. Although the functional characteristics of C3 in inflammation and phagocytosis have been elucidated at the gene expression level, it is better to confirm the C3 function at the protein level. As the expression level between proteins and mRNAs does not always show a simple linear relationship, follow-up studies at the protein level need further exploration.

In conclusion, the current study clarified that C3 has a conserved structure and function of host defense in Nile tilapia. As schematically illustrated in Figure 7, OnC3 is involved in the processes of relieving inflammation, reducing tissue damage, and promoting MO/Mø phagocytosis. These data further support that OnC3 is related to antibacterial immunity. Overall, the findings of this research suggest that C3 plays an important role in innate immunity, which helps to further refine the function and mechanism of action of teleost C3.

## 4. Materials and Methods

### 4.1. Animals and Sample Acquisition

Nile tilapia used in this study were obtained from the Guangdong Tilapia Breeding Farm [45,55]. The weight of fish was 70–100 g for tissue collection and 250–300 g for the isolation of Nile tilapia MO/Mø [45,55]. The Nile tilapia were cultured in a constant temperature semi-automatic circulatory system in the School of Life Sciences, South China Normal University for at least three weeks. The water temperature was stabilized at about 25–26 °C, pH was controlled approximately at 7.0, and the visibility was dominated at 25–30 cm. The oxygen solubility, nitrite contents, along with hydrogen sulfide were detected and adjusted regularly. In all experimental processes, the regulation of the South China Normal University Institutional Animal Care and Use Committee (reference number 2019-02-0016), as well as the ethics of medical laboratory animals were strictly adhered [45,55].

Challenges and sampling methods refer to published articles, and the brief steps are as follows [56]. During the experiments, the Nile tilapia were anesthetized with 40 mg/L MS-222 (Aladdin, Los Angeles, CA, USA). In this study, liver, brain, head kidney, thymus, spleen, skin, intestine, hind kidney, peripheral blood, and muscle were collected for tissue distribution patterns while only liver, spleen, and head kidney were acquired in challenge experiments. The collected tissues of each group were quick-frozen in liquid nitrogen and then transferred to −80 °C for cryopreservation. The challenge groups were stimulated by intraperitoneal injection of 100 µL live *S. agalactiae* or *A. hydrophila* (1 × 10^7^ CFU/mL), while the control group was treated with 100 µL 10 mM sterile PBS. Three Nile tilapia were selected for each group of experiments to obtain samples. The time of obtaining sample was 0 h, 3 h, 6 h, 12 h, 24 h, 48 h, and 72 h after challenge.

### 4.2. Total RNA Extraction

The total RNA of Nile tilapia tissue and cell samples were extracted with Trizol Reagent (Thermo, Waltham, MA, USA) according to the instructions. The tissue samples mixed with 200 µL Trizol were ground to homogeneity with an electric homogenizer, and then the Trizol was added to a final volume of 1 mL. The mixture was then placed on ice for 10 min and centrifuged at 12,000 rpm for 15 min at 4 °C. Then, the supernatant was mixed with 450 µL of chloroform in a new centrifuge tube. The above procedure was then repeated for standing and centrifugation. The upper organic phase was mixed with 600 µL of isopropanol in a new centrifuge tube. After standing and centrifugation, the bottom RNA pellet was collected. RNA samples were then washed with 75% ethanol solution twice. The resulting total RNA was confirmed by 1% agarose gel electrophoresis and the RNA concentration was determined using a Nanodrop 2000 spectrophotometer (Thermo, USA). All RNA samples were stored at −80 °C until use [57].

### 4.3. cDNA Synthesis

The cDNA template synthesis was performed with Hifair^®^ II 1st Strand cDNA Synthesis SuperMix for qPCR (gDNA digester plus) (YEASEN, Shanghai, China), and the brief steps were as follows. The reaction contained 1 µg of total RNA, 2.0 µL of 5 × gDNA digestion buffer, and 1.0 µL of gDNA digestion, and these were finally added to a total volume of 10 µL with RNase-free ddH_2_O. The mixture was maintained at 42 °C for 2 min. Then, 2 µL primescript RT enzyme mix, 1 µL RT primer mix, and 2.0 µL 5 × Hifair^®^ II (YEASEN, Shanghai, China) Buffer plus were mixed with the reaction solution from the first step, then RNase-free ddH_2_O was added to bring the total volume to 20 µL. After that, the mixture was maintained at 25 °C for 5 min, 42 °C for 30 min, and 85 °C for 5 min. Prepared cDNA templates were stored at −20 °C until use. The cDNA template was diluted to a total volume of 200 µL with RNase-free ddH_2_O before being used for RT-qPCR.

### 4.4. Amplification and Identification of OnC3

The predicted OnC3 sequence from NCBI (https://www.ncbi.nlm.nih.gov/ (accessed on 19 November 2020)) was acquired with XP_019203045.1. The sequencing primers were designed with Primer Premier 5 based on the predicted sequence and synthesized by BGI, China. All primers used in this study are listed in Table 1. The *OnC3* ORF was confirmed by polymerase chain reaction (PCR). The PCR reaction volume was 25 µL, consisting of 12.5 µL LA Taq Mix (Takara, Japan), 7.5 µL ddH_2_O, 3 µL cDNA template, 1 µL forward primer, and 1 µL reverse primer. The PCR reaction process contained 5 steps, pre-denaturation (95 °C) for 5 min, denaturation (95 °C) for 1 min, annealing (50 °C) for 30 s, extension (72 °C) for 2 min, and final extension (72 °C) for 5 min. In those steps, denaturation, annealing, and extension were performed for 35 cycles. After amplification, the PCR product was detected with 1% agarose gel electrophoresis and sequenced by BGI, China.

### 4.5. Bioinformatics Analysis of OnC3

In this study, the multiple sequence alignment and identity of *C3* were carried out with Blast X. The nucleotide sequence of *OnC3* was translated into amino acid sequence with ExPASy tools (http://expasy.org/tools/ (accessed on 14 May 2021)) and the protein domain was analyzed with the Simple Modular Architecture Research Tool (SMART) (http://smart.embl-heidelberg.de/ (accessed on 14 May 2021)). The signal peptide was predicted with SignalP 4.1 (http://www.cbs.dtu.dk/services-/SignalP/ (accessed on 14 May 2021)). The phylogenetic tree was constructed with MEGA 7 based on the neighbor-joining (NJ) algorithm. The structure of OnC3 was predicted by the server equipped with A100 GPU. The Github source code from AlphaFold2.0 (https://github.com/deepmind/alphafold/ (accessed on 23 December 2021)) [58].

### 4.6. Real-Time Quantitative PCR

The expression of *OnC3* transcripts in the tissues was performed with QuantStudio 5 Real-time PCR system (ABI, Los Angeles, CA, USA). The volume of the reaction was 20 µL, including 10 µL Hieff^®^ qPCR SYBR Green Master Mix (YEASEN, China), 2 µL forward primers, 2 µL reverse primers, 3 µL cDNA template, and 4 µL ddH_2_O. The program was a two-step method comprising pre-denaturation (95 °C) for 5 min, denaturation (95 °C) for 10 s, and annealing/extension (60 °C) for 30 s. The denaturation and annealing/extension were performed for 40 cycles. The melt curve was acquired in the last stage of RT-qPCR. Nile tilapia *β*-actin was selected as the internal reference for calculating the relative expression of target genes based on the 2^−ΔΔCT^ method [59]. In this research, all experiments were repeated three times.

### 4.7. Isolation of Nile Tilapia Head Kidney MO/Mø

The isolation of Nile tilapia head kidney MO/Mø was developed based on the references [60,61]. The head kidneys were obtained from healthy fish and mashed into tissue homogenate in sterile dishes with RPMI 1640 medium. The cell suspensions were obtained and gently loaded into the separation liquid, which consisted of 10 mL 54% percoll (Sigma, St. Louis. MO, USA) blanketed with 10 mL of 31% percoll. After that, the separation liquid was centrifuged at 4 °C, 500× *g* for 40 min, then the MO/Mø fraction of the 31–54% interface was collected and washed three times with RPMI 1640 medium. The cell concentration was adjusted to 5 × 10^6^ cells/mL by the L-15 medium with 1% penicillin/streptomycin and 10% FBS. The isolated cells were cultured in 96-well plates (100 µL/well) (Thermo, USA) and incubated at 25 °C. After 3 days, the non-adherent cells were removed. The adherent Nile tilapia MO/Mø were resuspended and washed with 10 mL 1640 medium 3 times. The cells were diluted to 1 × 10^6^ cells/mL and incubated in 96-well plates for experiments.

In order to explore the expression pattern of *OnC3* in MO/Mø after bacterial challenge, the head kidney (1 × 10^6^ cells/mL) cells were challenged with formalin inactivated *S. agalactiae* or *A. hydrophila* (1 × 10^7^ CFU/mL) while the control was treated with sterile PBS (10 mM). After stimulation, the cell samples were collected at 3 h, 6 h, 12 h, 24 h, 48 h, and 72 h.

### 4.8. Expression and Purification of (r)OnC3

The *β* chain of *OnC3* was amplificated and inserted into the *E. coli* expression vector Trx-pET-32a (Trx). The recombinant Trx-OnC3 construct was transformed into *E. coli.* BL21 (DE3) competent cells and coated on LB medium plates consisting of 1% ampicillin. On the next day, the positive clones were selected, identified with PCR reaction, and sequenced by BGI, China. The cells were cultured with 250 mL LB-ampicillin medium until the OD value at 600 nm was 0.6–1.0. Then, the bacteria liquid was induced with 1 mM isopropyl *β*-D-thiogalactoside (IPTG) at 37 °C for 6 h. After that, the target recombinant protein was purified with His-Band Resin columns (Novagen, Darmstadt, Germany) according to the prot °Cols. Finally, the purified protein was detected by 10% SDS-PAGE. The Trx vector protein was prepared through the same methods for the follow-up experiments [45,55].

### 4.9. (r)OnC3 Regulates the Expression of Inflammation-Related Factors

In order to explore the effects of (r)OnC3 on inflammatory response in vivo, the experimental group was intraperitoneally injected with 100 µL (r)OnC3 protein (50 µg/mL), 100 µL Trx protein (50 µg/mL), and 100 µL PBS (10 mM). The liver and spleen of tilapia were detected at 12 h after stimulation. Sample collection and template preparation are referred to Section 4.1, Section 4.2 and Section 4.3. The mRNA expressions of pro-inflammatory cytokines *IL-1β*, *IL-6,* and *TNF-α* and anti-inflammatory cytokines *IL-10* and *TGF-β* were analyzed with RT-qPCR based on the methods in Section 4.6 [62]. In addition, the head kidney MO/Mø were isolated according to Section 4.7 and treated with (r)OnC3 protein (50 µg/mL), Trx protein (50 µg/mL), or sterile PBS (10 mM). The (r)OnC3 protein as well as Trx protein were used after filter sterilization. Samples were collected at 12 h, 24 h, 48 h, and 72 h post-stimulation for RNA extraction and RT-qPCR analysis.

Moreover, the regulation of inflammation by (r)OnC3 after pathogen infection was also examined. Nile tilapia were intraperitoneally injected with 100 µL sterile PBS (10 mM), *S. agalactiae* (1 × 10^7^ CFU/mL), *S. agalactiae* + Trx protein (50 µg/mL), or *S. agalactiae* + (r)OnC3 protein (50 µg/mL). The liver and spleen were harvested 12 hours after stimulation, and *IL-1β*, *IL-6*, *TNF-α*, *IL-10,* and *TGF-β* were detected. The isolation and culture methods of MO/Mø were the same as in Section 4.7. The head kidney (1 × 10^6^ cells/mL) MO/Mø were treated with sterile PBS (10 mM), *S. agalactiae* (1 × 10^7^ CFU/mL), *S. agalactiae* + Trx protein (50 µg/mL), or *S. agalactiae* + (r)OnC3 protein (50 µg/mL), and the cells were collected for RNA extraction after 12 h. The same inflammatory factors were detected by RT-qPCR.

### 4.10. Histopathological Analysis

All in vivo stressed fish livers and spleens in Section 4.9 were taken in duplicate, one for RNA extraction and one for pathological examination. Histopathological section preparation and analysis were performed according to publications [45]. The obtained Nile tilapia liver and spleen were immediately fixed with 4% paraformaldehyde solution at 4 °C overnight. The fixed samples were sent to Servicebio Biological Technology Co., Ltd. (Wuhan, China), for histopathological analysis by H&E staining. Three Nile tilapia were randomly selected from each group for pathological statistical analysis. Scoring was performed by two blinded investigators. The liver and spleen injury score were recorded via a scale of 0 to 3 (normal 0, mild 1, moderate 2, severe 3) according to the inflammatory cell infiltration, cell necrosis, and tissue vacuolation [63,64].

### 4.11. Determination of ROS in MO/Mø

In this research, MO/Mø were isolated and cultured using the method in Section 4.7. The detection of reactive oxygen species (ROS) was carried out by the ROS Assay Kit of Beyotime Biotechnology [65]. The DCFH-DA was diluted with serum-free culture medium at 1:1000 to make the final concentration 10 µmol/L. After removing the cell culture medium, the cells (1 × 10^7^ cells/mL) were treated with 200 µL DCFH-DA and incubated at 37 °C for 20 min. Cells were washed three times by serum-free cell culture medium. The MO/Mø in the experimental group was treated with 200 µL (r)OnC3 protein (50 µg/mL). The other groups were treated with 200 µL sterile PBS (10 mM) or Trx protein (50 µg/mL). After 30 min of stimulation at 37 °C, the DCF fluorescence was detected by FACS Aria III (BD, New York, NY, USA) flow cytometer with an argon-ion laser of 488 nm. A total of 10,000 individual cells were analyzed in each of the samples.

### 4.12. Detection of Phagocytosis-Related Receptors and Enzyme Expression

In Section 4.9, except for the liver and spleen, head kidney tissue was taken from all in vivo stimulation groups and stored at −80 °C according to the previous steps. Similarly, the cell samples were stimulated and collected as in Section 4.9. The mRNA expressions of phagocytosis-related receptors (*CD11b*, *CD18*), enzymes (*RhoA, ROCK*), and lysosome-related factors (*EEA*, *M6PR,* and *vATPase*) in the head kidney and head kidney MO/Mø were detected at time of 6 h after challenge. The RNA extraction, cDNA synthesis, and RT-qPCR detection were performed according to the above sections.

### 4.13. Assay of Nile Tilapia Head Kidney MO/Mø Phagocytosis

Before experiments, the Nile tilapia head kidney MO/Mø and *S. agalactiae* (1 × 10^8^ CFU/mL) labeled with fluorescein isothiocyanate (FITC) (Sigma, USA) were prepared while the forward scatter (FSC) and side scatter (SSC) were measured [56,57]. After that, 100 µL Trx (50 µg/mL) or (r)OnC3 (50 µg/mL) were added to 300 µL labeled bacteria and incubated avoid light for 1 h. The control group was treated with 100 µL sterile PBS (10 mM). Then, the suspensions were centrifuged, and the bacteria were resuspended with 300 µL PBS and 300 µL cell suspension (2 × 10^7^ cells/mL) and incubated for 60 min with shaking. The solution above was centrifuged with over 3% BSA in PBS supplemented at 100× *g*, 4 °C for 10 min to eliminate the non-ingested bacteria. The extra fluorescence was quenched by mixing 1 µL trypan blue (0.4%). Finally, the cell suspension was detected by FACS Aria III (BD, USA) flow cytometer with a 488 nm argon-ion laser [61]. A total of 10,000 individual cells were analyzed in each of the samples.

### 4.14. Statistical Analysis

All data in the present research were repeated three times and shown as mean ± standard deviation (SD). Then, the data were analyzed by one-way ANOVA with SPSS 17.0. The statistical significance was indicated by letters and asterisks (* 0.01 ≤ *p* < 0.05, ** *p* < 0.01). The figures were made with GraphPad Prism 7 (GraphPad, San Diego, CA, USA).

## Figures and Tables

**Figure 1 ijms-23-15586-f001:**
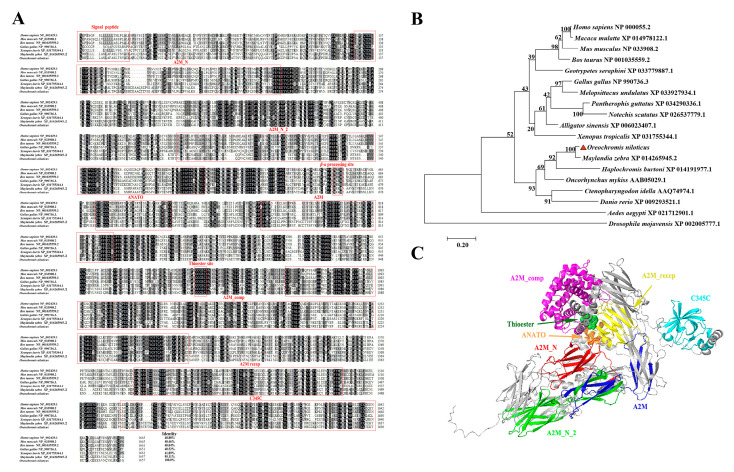
Sequence analysis of OnC3 and 3D structure diagram of OnC3 protein. (**A**) Multiple sequence alignment of the deduced amino acid sequences of C3 among different species. GenBank accession numbers of the used species for comparison are *Homo sapiens* (NP_000055.2), *Mus musculus* (NP_033908.2), *Bos taurus* (NP_001035559.2), *Gallus* (NP_990736.3), *Xenopus laevis* (XP_031755344.1), and *Maylandia zebra* (XP_014265945.2), respectively. The conservative motifs were highlighted in red box, including important domains such as A2M_N, A2M_N_2, ANATO, A2M, A2M_comp, A2M_recep, and C345C. (**B**) Phylogenetic tree of C3 family members constructed using the NJ method by MEGA 7.0 program based on the alignment of 19 members performed with the Clustal W method. Numbers at each branch indicate the percentage bootstrap values on 1000 replicates. The red triangle represents *O. niloticus* C3 gene cloned in this study. (**C**) The 3D structure diagram of OnC3 protein. Alphfold was used for protein structure prediction, and Pymol was used for software drawing. Different domains of OnC3 protein are distinguished by different colors.

**Figure 2 ijms-23-15586-f002:**
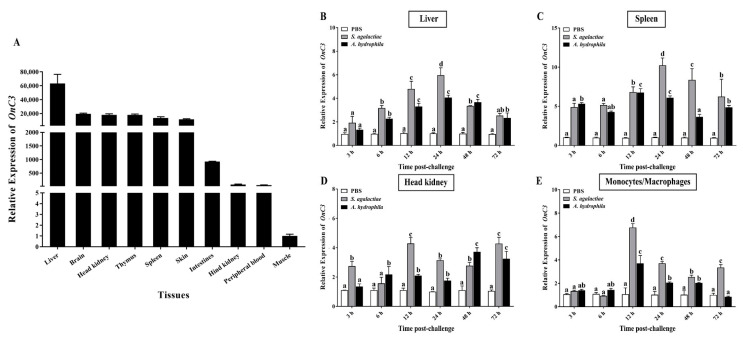
(**A**) Tissue distribution pattern of *OnC3* transcript in healthy Nile tilapia. The expression of *OnC3* in different tissues is compared to that in the muscle and normalized against the expression of *β*-actin. (**B**–**D**) Expression of *OnC3* transcript in the *S. agalactiae* (1 × 10^7^ CFU/mL) or *A. hydrophila* (1 × 10^7^ CFU/mL) challenged tissues (liver, spleen, and head kidney). The expression of *OnC3* was normalized to *β*-actin, and the multiple changes in *OnC3* in each tissue were calculated deciding that in the PBS-immunized group. (**E**) Expression of *OnC3* in the MO/Mø after stimulation with inactivated *S. agalactiae* (1 × 10^7^ CFU/mL) and *A. hydrophila* (1 × 10^7^ CFU/mL). The error bars represent standard deviation (*n* = 3), and the significant differences are indicated by different letters (a, b, c, d) (*p* < 0.05).

**Figure 3 ijms-23-15586-f003:**
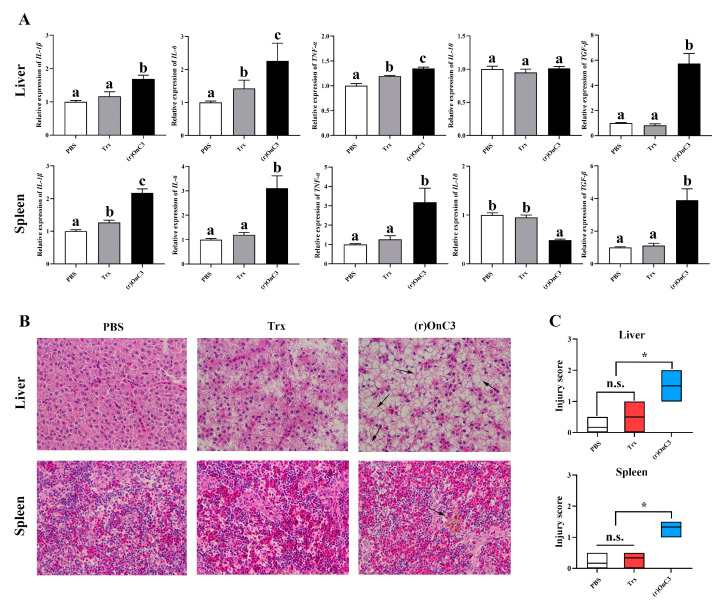
The effects of (r)OnC3 on tilapia inflammatory response in vivo. (**A**) The mRNA expressions of inflammatory cytokines *IL-1β*, *IL-6,* and *TNF-α* and anti-inflammatory cytokines *IL-10* and *TGF-β* in the liver and spleen of tilapia were detected at 12 h after treated with (r)OnC3 protein (50 µg/mL). The control groups were treated with 10 mM sterile PBS and Trx protein (50 µg/mL). (**B**) (r)OnC3 promoted inflammatory infiltration in the liver and spleen by H&E staining (×400). (**C**) The injury score of the liver and spleen. The error bars represent standard deviation (*n* = 3), and the significant differences are indicated by different letters (a, b, c) (*p* < 0.05) and asterisks (n.s. 0.05 < *p*, * 0.01 ≤ *p* < 0.05).

**Figure 4 ijms-23-15586-f004:**
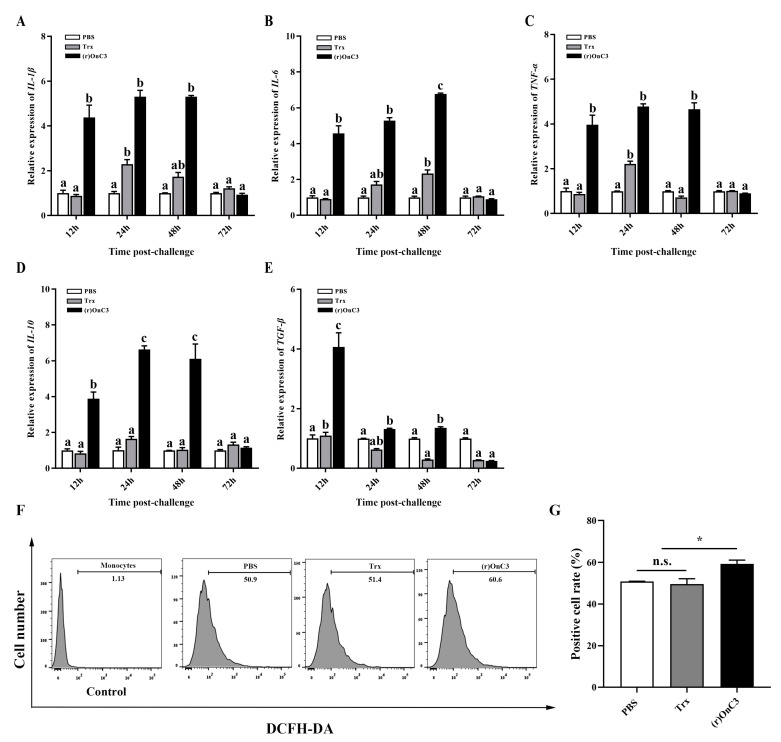
The effects of (r)OnC3 on tilapia MO/Mø inflammatory response and ROS in vitro. (**A**–**E**) The mRNA expressions of inflammatory cytokines *IL-1β*, *IL-6,* and *TNF-α* and anti-inflammatory cytokines *IL-10* and *TGF-β* were detected from Nile tilapia MO/Mø at 12 h, 24 h, 48 h, and 72 h after treatment with (r)OnC3 protein (50 µg/mL). The control groups were treated with 10 mM sterile PBS and Trx protein (50 µg/mL). (**F**) Flow cytometric analysis of the MO/Mø ROS preincubated with PBS, Trx, or (r)OnC3 (50 µg/mL). The histogram result shows the DCFH-DA intensity of the circled myeloid gate cells, and 10,000 cells were analyzed. (**G**) The percentage of positive head kidney MO/Mø was analyzed by flow cytometry. The error bars represent standard deviation (*n* = 3), and the significant differences are indicated by different letters (a, b, c) (*p* < 0.05) and asterisks (n.s. 0.05 < *p*, * 0.01 ≤ *p* < 0.05).

**Figure 5 ijms-23-15586-f005:**
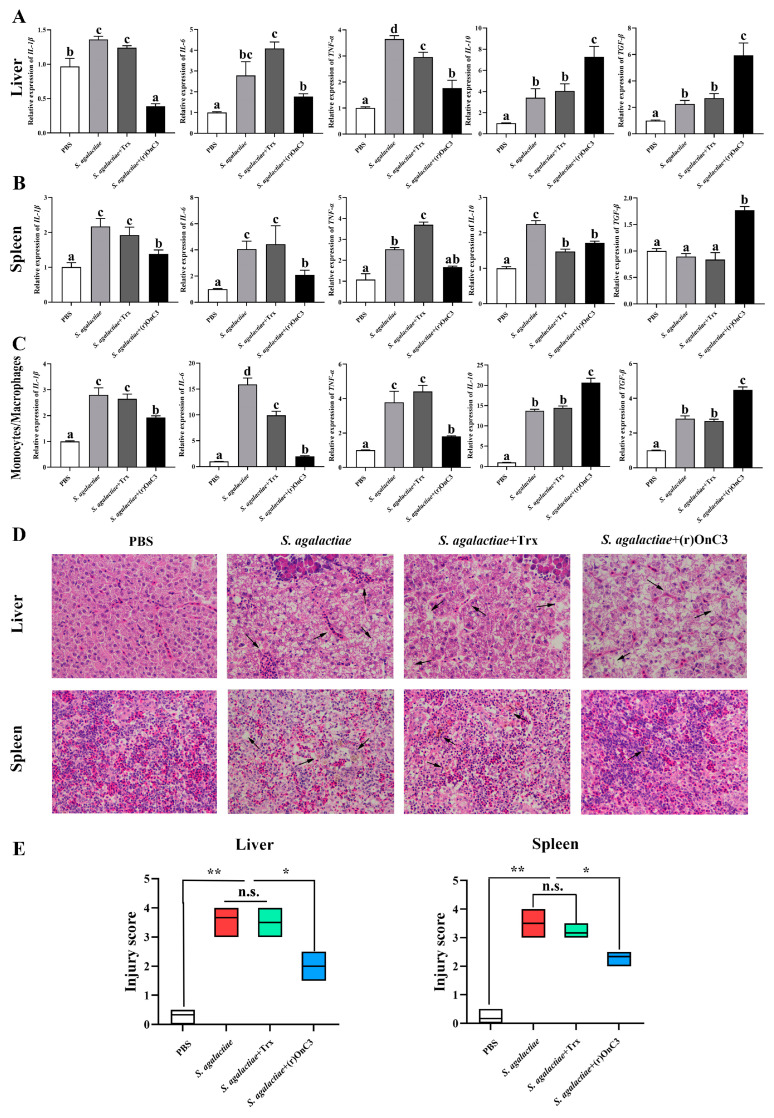
The effects of (r)OnC3 on inflammation after Nile tilapia infected with *S. agalactiae* (1 × 10^7^ CFU/mL) in vivo and in vitro. (**A**) The mRNA expressions of inflammatory cytokines *IL-6*, *IL-8,* and *TNF-α* and anti-inflammatory cytokines *IL-10* and *TGF-β* were detected from Nile tilapia liver at 12 h after challenge with (r)OnC3 protein and *S. agalactiae.* The control group was treated with 10 mM sterile PBS, and the rest of the groups were treated with *S. agalactiae* (1 × 10^7^ CFU/mL), *S. agalactiae* + Trx protein (50 µg/mL), or *S. agalactiae* + (r)OnC3 protein (50 µg/mL). (**B**) The same inflammatory factors were detected in spleen, and the experimental conditions were the same as those in panel A. (**C**) The same inflammatory factors were detected in MO/Mø, and the experimental conditions were the same as those in panel A. (**D**) (r)OnC3 inhibited inflammatory infiltration in the bacterial-infected liver and spleen by H&E staining (×400). The control group was challenged with PBS, and the positive groups were challenged with *S. agalactiae* and *S. agalactiae* with Trx protein, respectively. The experiment groups were challenged with *S. agalactiae* with (r)OnC3 protein. (**E**) The injury score of liver and spleen. The error bars represent standard deviation (*n* = 3), and the significant differences are indicated by different letters (a, b, c, d) (*p* < 0.05) and asterisks (n.s. 0.05 < *p*, * 0.01 ≤ *p* < 0.05, ** *p* < 0.01).

**Figure 6 ijms-23-15586-f006:**
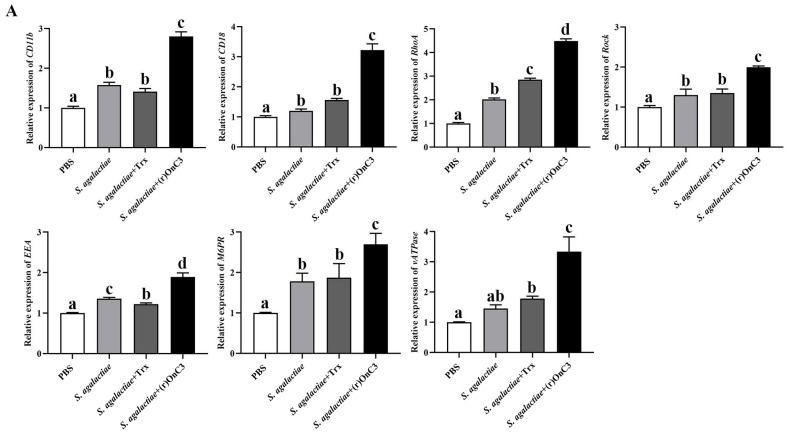
The regulatory effect of (r)OnC3 on Nile tilapia MO/Mø phagocytosis in vivo and in vitro. (**A**) The mRNA expressions of phagocytosis-related receptors and enzymes *CD11b*, *CD18*, *RhoA,* and *Rock* and lysosome-related genes *EEA*, *M6PR,* and *vATPase* were detected from Nile tilapia head kidney tissue at 6 h after treated with (r)OnC3 protein. The control group was treated with 10 mM sterile PBS, the rest of the groups were treated with *S. agalactiae* (1 × 10^7^ CFU/mL), *S. agalactiae* + Trx (50 µg/mL) protein, or *S. agalactiae* + (r)OnC3 protein (50 µg/mL). (**B**) The mRNA expression of phagocytosis-related receptors and enzymes *CD11b*, *CD18*, *RhoA,* and *ROCK* and lysosome-related genes *EEA*, *M6PR,* and *vATPase* were detected from Nile tilapia MO/Mø at 6 h after treated with (r)OnC3 protein. The blank control group was treated with 10 mM sterile PBS, the rest of the groups were treated with *S. agalactiae* (1 × 10^7^ CFU/mL), *S. agalactiae* + Trx protein (50 µg/mL), or *S. agalactiae* + (r)OnC3 (50 µg/mL) protein. (**C**) The phagocytic rate of MO/Mø to *S. agalactiae* (1 × 10^7^ CFU/mL) was determined by flow cytometry. The MO/Mø phagocytosing *S. agalactiae* was preincubated with PBS, Trx protein, or (r)OnC3 protein, and the phagocytosis rates are shown near the marker. The histogram result shows the FITC intensity of the circled myeloid gate cells, and 10,000 cells were analyzed. (**D**) The phagocytosis rates. The error bars represent standard deviation (*n* = 3), and significant difference was indicated by different letters (a, b, c, d) (*p* < 0.05) and asterisks (n.s. 0.05 < *p*, * 0.01 ≤ *p* < 0.05).

**Figure 7 ijms-23-15586-f007:**
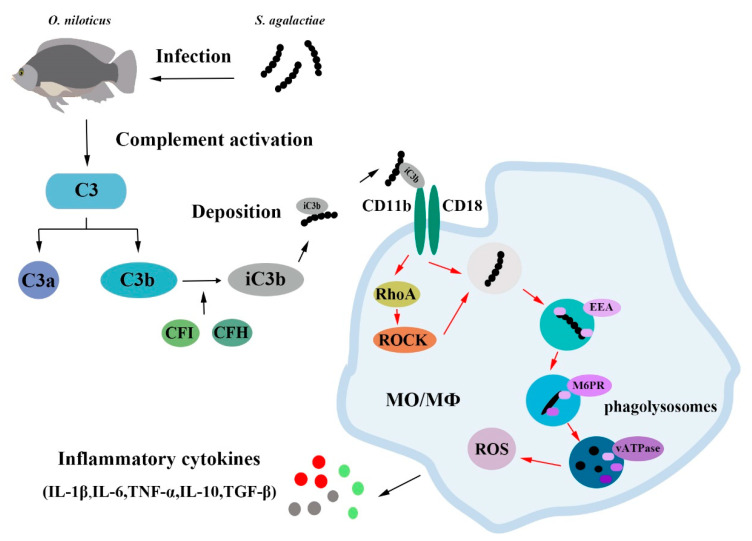
Complement C3 regulates inflammatory response and monocyte/macrophage phagocytosis to *Streptococcus agalactiae* in a teleost fish. The proposed model illustrates the mechanism by which OnC3 as a core molecule of the complement system effectively participates in non-specific cell immunity. After *S. agalactiae* infects fish, the C3 molecule is decomposed into C3a and C3b, and the inactivated iC3b can combine with CR3 (CD11b/CD18) to enhance the phagocytosis of monocytes/macrophages. The C3 molecule also modulates the inflammatory response to defend against *S. agalactiae* infection.

**Table 1 ijms-23-15586-t001:** The primers used in this study.

Primers	Sequence (5′–3′)	Purpose
OnC3-F1	TGTCCCCTTCGTCATTATTCC	Sequencing
OnC3-R1	CATCTCCTCCAAGCCCAAGC	Sequencing
OnC3-F2	GGAGGGAGGTTAGTGAATAAAAGAA	Sequencing
OnC3-R2	GGAATAATGACGAAGGGGAC	Sequencing
OnC3-F3	AAAGGACCAAAGCAAGC	Sequencing
OnC3-R3	GGCTATGATGCGGAACGATG	Sequencing
EOnC3-F	CCGGAATTCAACCATCCAACCAAAACGA	Protein expression
EonC3-R	CCCAAGCTTTATGATGCGGAACGATGG	Protein expression
qC3-F	GAGCGGGACATCAACAGCC	RT-qPCR
qC3-R	CTCAGTTCAGCCTCCACCATTT	RT-qPCR
M13-F	TGTAAAACGACGGCCAGT	Sequencing
M13-R	CAGGAAACAGCTATGACC	Sequencing
T7	TAATACGACTCACTATAGGG	Sequencing
T7t	GCTAGTTATTGCTCAGCGG	Sequencing
*β*-actin-F	CGAGAGGGAAATCGTGCGTGACA	Control, RT-qPCR
*β*-actin-R	AGGAAGGAAGGCTGGAAGAGGGC	Control, RT-qPCR
qIL-1*β*-F	GTTCACCAGCAGGGATGAGATT	RT-qPCR
qIL-1*β*-R	TGCGGTCTTCACTGCCTCC	RT-qPCR
qIL-6-F	ACAGAGGAGGCGGAGATG	RT-qPCR
qIL-6-R	GCAGTGCTTCGGGATAGAG	RT-qPCR
qTNF-*α*-F	GCTGAGGCTCCTGGACAAAA	RT-qPCR
qTNF-*α*-R	TCTGCCATTCCACTGAGGTCTT	RT-qPCR
qIL-10-F	TGGAGGGCTTCCCCGTCAG	RT-qPCR
qIL-10-R	CTGTCGGCAGAACCGTGTCC	RT-qPCR
qTGF-*β*-F	CAAACACGCTGAAGGACAAATG	RT-qPCR
qTGF-*β*-R	CGTTATTGCCGCATTCACAG	RT-qPCR
qCD11b-F	GGAAATGATGAAGAACCAGGTGTC	RT-qPCR
qCD11b-R	GCAGCCTTGCCGTGAATGTG	RT-qPCR
qCD18-F	GAGTCGTCTACAGGCCGATATGCA	RT-qPCR
qCD18-R	AATGCCCCGTGGACTGATGGT	RT-qPCR
qRhoA-F	ACAGGATTGGTGCCTTTGG	RT-qPCR
qRhoA-R	AGTAGGACGCATTTATTGCTCTT	RT-qPCR
qROCK-F	TTGGAGGGCTGGCTGTCTA	RT-qPCR
qROCK-R	CTTCTTACTGCTCACCACCACAT	RT-qPCR
qEEA-F	GGTGCGTGAAGGGTGAAGGT	RT-qPCR
qEEA-R	ATCTGAAGCGACTGGTTCTCTCTGC	RT-qPCR
qM6PR-F	GCAATAAGGAAGAAAGGAAAGCC	RT-qPCR
qM6PR-R	TCCAGAATCCCCTCCAGCGT	RT-qPCR
qvATPase-F	CGAGACCTCAAGTGGGAGTTTT	RT-qPCR
qvATPase-R	CATACCATAGATGTCGCCGCC	RT-qPCR

## Data Availability

The data presented in this study are available from the corresponding author on reasonable request.

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
