# Peer review of "Complement C3 Regulates Inflammatory Response and Monocyte/Macrophage Phagocytosis of Streptococcus agalactiae in a Teleost Fish"

_ijms, 2022, doi:10.3390/ijms232415586_

Round 1
Reviewer 1 Report
In this study by Bai et al, C3 from Nile tilapia was cloned and its function in resisting pathogen infection was characterized. Data in this study provide a theoretical reference for in-depth understanding of C3 in host defense against bacterial infection and the immunomodulatory roles in teleost fish. This study is well designed and the conclusion is supported by its results. I have a few minor concerns about the study which are mentioned below.
1. Why did the authors select Nile tilapia for their study? Please provide the rationale for using this model.
2. There are many grammatical errors throughout the manuscript.
3. Why tissues e.g., intestine, brain, muscle, etc were not acquired in challenged experiments?
Author Response
Response to reviewers.
Comments to the Author (in italics) with our responses (bold script)
We believe that we have accommodated the requests of the reviewers and in doing so, the quality of the manuscript has been greatly improved.
Reviewer #1
In this study by Bai et al, C3 from Nile tilapia was cloned and its function in resisting pathogen infection was characterized. Data in this study provide a theoretical reference for in-depth understanding of C3 in host defense against bacterial infection and the immunomodulatory roles in teleost fish. This study is well designed and the conclusion is supported by its results. I have a few minor concerns about the study which are mentioned below.
Minor issues
- Why did the authors select Nile tilapia for their study? Please provide the rationale for using this model.
We appreciate the reviewer’s comment. Nile tilapia (Oreochromis Niloticus) is a teleost fish with a wide distribution range, high nutritional value, and strong environmental adaptability [1, 2]. Nile tilapia is not only an important economically farmed fish, but also an important research model for bony fish [3]. Nile tilapia has become a fish species recommended by the Food and Agriculture Organization of the United Nations and is currently ranked as the second largest farmed freshwater fish in global production [4, 5]. While, in recent years, Streptococcus agalactiae has caused high mortalities of tilapia in China, which restricts the development of the tilapia aquaculture industry [6]. Until now, research on the mechanism of tilapia immune defense against pathogenic microorganisms is still limited [7, 8]. Thus, Nile tilapia was chosen as the fish model in this study.
Following the reviewer's valuable suggestion, the description in the introduction has been modified in line 82-84 in the revised manuscript.
Line 82-84 (Supplemented in the revised manuscript): Therefore, further research on the mechanisms of Nile tilapia immune defense against pathogenic microorganisms is an urgent need for green aquaculture [8, 30-32].
- Bavia L, Santiesteban-Lores LE, Carneiro MC, Prodocimo MM. Advances in the complement system of a teleost fish, Oreochromis niloticus. Fish Shellfish Immun. (2022) 123:61-74.
- Liang Z, Li L, Fitzsimmons K. Tilapia germplasm in China: chance and challenge. In: International Symposium on Better Science, Better Fish (2011).
- Rey AL, Asín J, Zarzuela IR, et al. A proposal of standardization for histopathological lesions to characterize fish diseases. Rev. Aquacult. (2020) (3): 2304-2315.
- FAO, The state of world fisheries and aquaculture 2020, in: Sustainability in Action, FAO, 2020.
- Costa-Pierce BA. Rapid evolution of an established feral tilapia (Oreochromis spp.): the need to incorporate invasion science into regulatory structures, Biol. Invasions (2003) 5: 71-84.
- Ewees AA, Hemedan AA., Hassanien AE. Sahlol AT. Optimized support vector machines for unveiling mortality incidence in tilapia fish, Ain. Shams. Eng. (2021) 12.
- Pretto-Giordano LG, Müller EE, De Freitas JC, Da Silva VG, Evaluation on the pathogenesis of Streptococcus agalactiae in Nile tilapia (Oreochromis niloticus), Braz. Arch. Biol. Technol. (2010) 53: 87-92
- Yin X, Wu H, Mu L, et al. Identification and characterization of Calreticulin (CRT) from Nile tilapia (Oreochromis niloticus) in response to bacterial infection. Aquaculture (2020) 529: 73570.
- There are many grammatical errors throughout the manuscript.
We appreciate the reviewer’s comment. We carefully checked the full text for grammatical problems and corrected them in the manuscript. The revised sections have been modified in the manuscript using the “Track Changes” function.
Line 36: “The activation of complement is achieved through the classical pathway”
Line 81: “…causes serious damage to Nile tilapia tissues…”
Line 99: “The phylogenetic tree showed”
Line 165-166: “The expression of TGF-β in the (r)OnC3 group also was 5-fold higher than in other groups at 12 h after stimulation”
Line 257-258: “C3 and its cleavage fragments bind to receptors on the cell surface”
Line 331-332: “Recent studies pointed out that C3a enhances the phagocytic activity of B cells”
Line 337-338: “It can bind to inactivated C3b (iC3b) and activate phagocytosis…”
Line 338: “Despite the signaling pathway(s)”
Line 358-359: “…OnC3 is related to antibacterial immunity”
Line 364: “The proposed model illustrates…”
Line 449-451: “Nile tilapia β-actin was selected as internal reference for calculating the relative expression of target genes based on the 2-ΔΔCT method”
Line 487-488: Sample collection and template preparation are referred to Sections 2.1-2.3.
Line 521-522: “…200 μL DCFH-DA and incubated at 37 oC for 20 min”
Line 524: “After 30 min of stimulation at 37oC”
Line 549-550: “All data in the present research were repeated three times and shown as mean ± standard deviation (SD).”
- Why tissues e.g., intestine, brain, muscle, etc were not acquired in challenged experiments?
We appreciate the reviewer’s comment. According to the tissue distribution presented in Fig. 2A, the expression of OnC3 mRNA in liver, brain, head kidney, thymus and spleen at a high level. According to previous reports, liver, spleen and head kidney are important immune tissues of Nile tilapia [1, 2]. These organs are the main targets that are attacked by bacteria after intraperitoneal injection and mobilizes the circulation of lymphocytes through the body [3-5]. Hepatocytes and macrophages in the head kidney and spleen are the main cells producing complement molecules [6, 7]. Collectively, the tissues of liver, spleen, and head kidney were chosen in the challenge experiments.
- Magnadottir B. Innate immunity of fish (Overview). Fish Shellfish Immun. (2006) 20: 137-51.
- Alvarez-Pellitero P. Fish immunity and parasite infections: from innate immunity to immunoprophylactic prospects. Vet. Immunol. Immunopathol. (2008) 126: 171-198.
- Mcintire CR, Yeretssian G, Saleh M. Inflammasomes in infection and inflammation. Apoptosis (2009) 14: 522-35.
- Mu L, Yin X, Wu H, et al. Mannose-binding lectin possesses agglutination activity and promotes opsonophagocytosis of macrophages with Calreticulin interaction in an early vertebrate. J. Immunol. (2020) 205: 3443-55.
- Yin X, Li X, Chen N, et al. Hemopexin as an acute phase protein regulates the inflammatory response against bacterial infection of Nile tilapia (Oreochromis niloticus). Int. J. Biol. Macromol. (2021) 187: 166-78.
- Singer L, Colten HR, Wetsel RA. Complement C3 deficiency: Human, animal, and experimental models. Pathobiology (1994) 62:14-28.
- Bavia L, Santiesteban-Lores LE, Carneiro MC, Prodocimo MM. Advances in the complement system of a teleost fish, Oreochromis niloticus. Fish Shellfish Immun. (2022) 123: 61-74.

Reviewer 2 Report
Thank you very much for the chance to review this manuscript. The authors investigated the role of C3 in pathogen infections. The study is well conducted and presented in an intelligent fashion. I only have some minor comments.
-In Figure2B (24h), C (24h), and E (12h), the letter "d "seems to be incorrect as the legend states, "significant differences are indicated by different letters (a,b,c)"
Legend also states „…and asterisks (* 0.01 ≤ p < 0.05, ** p < 0.01).“
however, the figure doesn't contain asterisks.
-it would be good to include the limitations of your study in the discussion section. It would also be good to state that you mainly focussed on gene expression levels and not on the protein level (e.g., ELISA, Western blot) and that gene expression findings might not be found on the protein level in all cases.
Author Response
Response to reviewers.
Comments to the Author (in italics) with our responses (bold script)
We believe that we have accommodated the requests of the reviewers and in doing so, the quality of the manuscript has been greatly improved.
Reviewer #2
Thank you very much for the chance to review this manuscript. The authors investigated the role of C3 in pathogen infections. The study is well conducted and presented in an intelligent fashion. I only have some minor comments.
Minor issues
- In Figure2B (24h), C (24h), and E (12h), the letter "d "seems to be incorrect as the legend states, "significant differences are indicated by different letters (a,b,c)" Legend also states, "…and asterisks (* 0.01 ≤ p < 0.05, ** p < 0.01) " however, the figure does not contain asterisks.
We appreciate the reviewer’s valuable suggestion. The inappropriate interpretation has been corrected in line 142-143 in the revised manuscript.
Line 142-143: “…by different letters (a, b, c, d) (p < 0.05)”
Following the reviewer's suggestion, the similar errors have also been corrected in line 184-185, line 224 and line 253 in the revised manuscript.
Line 184-185: “…asterisks (* 0.01 ≤ p < 0.05).”
Line 224: “…different letters (a, b, c, d) (p < 0.05)”
Line 253: “…by different letters (a, b, c, d) (p < 0.05) and asterisks (* 0.01 ≤ p < 0.05)”
- It would be good to include the limitations of your study in the discussion section. It would also be good to state that you mainly focused on gene expression levels and not on the protein level (e.g., ELISA, Western blot) and that gene expression findings might not be found on the protein level in all cases.
We appreciate the reviewer’s helpful suggestion. According to the reviewer’s suggestion, the discussion has been modified in line 349-354 in the revised manuscript.
Line 349-354 (Supplemented in the revised manuscript): Although the functional characteristics of C3 in inflammation and phagocytosis have been elucidated at the gene expression level, it is better to confirm the C3 function at the protein level. As the expression level between proteins and mRNAs does not always show a simple linear relationship [1], follow-up studies at the protein level need further exploration.
- Buccitelli C, Selbach M. mRNAs, proteins and the emerging principles of gene expression control. Nat. Rev. Genet. (2020) 21:630-644.
